# Rationally Improving Doramectin Production in Industrial *Streptomyces avermitilis* Strains

**DOI:** 10.3390/bioengineering10060739

**Published:** 2023-06-20

**Authors:** Fujun Dang, Qingyu Xu, Zhongjun Qin, Haiyang Xia

**Affiliations:** 1Key Laboratory of Synthetic Biology, The Center of Excellent Plant Molecular Sciences, The Chinese Academy of Sciences, Shanghai 200032, China; 2Institute of Biopharmaceuticals, Taizhou University, Taizhou 317000, China

**Keywords:** *Streptomyces avermitilis*, doramectin, polyketide biosynthetic cluster, *fadD17*

## Abstract

Avermectins (AVMs), a family of 16-membered macrocyclic macrolides produced by *Streptomyces avermitilis*, have been the most successful microbial natural antiparasitic agents in recent decades. Doramectin, an AVM derivative produced by *S. avermitilis bkd^−^* mutants through cyclohexanecarboxylic acid (CHC) feeding, was commercialized as a veterinary antiparasitic drug by Pfizer Inc. Our previous results show that the production of avermectin and actinorhodin was affected by several other polyketide biosynthetic gene clusters in *S. avermitilis* and *Streptomyces coelicolor*, respectively. Thus, here, we propose a rational strategy to improve doramectin production via the termination of competing polyketide biosynthetic pathways combined with the overexpression of CoA ligase, providing precursors for polyketide biosynthesis. *fadD17*, an annotated putative cyclohex-1-ene-1-carboxylate:CoA ligase-encoding gene, was proven to be involved in the biosynthesis of doramectin. By sequentially removing three PKS (polyketide synthase) gene clusters and overexpressing FadD17 in the strain DM203, the resulting strain DM223 produced approximately 723 mg/L of doramectin in flasks, which was approximately 260% that of the original strain DM203 (approximately 280 mg/L). To summarize, our work demonstrates a novel viable approach to engineer doramectin overproducers, which might contribute to the reduction in the cost of this valuable compound in the future.

## 1. Introduction

Avermectins (AVMs) are a series of 16-membered macrocyclic lactones produced by *Streptomyces avermitilis* isolated from a soil sample collected in Shizuoka Prefecture, Japan, in 1978 (Figure 1A) [1]. AVMs and their derivatives were developed as the most successful antiparasitic compounds to prevent endo- and ectoparasitic infections of livestock and control important agricultural insects [2]. Because derivatives of avermectins (ivermectin) lowered the incidence of river blindness and other parasitic diseases, William C. Campbell and Satoshi Ōmura, the discoverers of avermectin, won the 2015 Nobel Prize in Physiology or Medicine together with the discovery of artemisinin [3,4]. Currently, avermectin and its derivative emamectin are the most popular natural pesticides worldwide [3].

Doramectin, which is sold commercially as Dectomax^®^, was produced by mutants of *S. avermitilis* lacking both branched-chain-2-oxo acid dehydrogenase (Bkd) and 5-*O*-methyltransferase (AveD) via feeding with cyclohexanecarboxylic acid (Figure 1A) [5,6]. Doramectin was already approved by the Food and Drug Administration (FDA) as a veterinary drug for the treatment of parasites such as gastrointestinal roundworms, lungworms, eyeworms, grubs, sucking lice, and mange mites in cattle, sheep, swine, and others [7]. It showed a longer plasma half-life and better persistent activity against the *Ostertagia ostertagi* and *Cooperia oncophora* infection of cattle [7,8].

Efforts to improve avermectin production have never stopped since the discovery of avermectin via classic mutagenesis and molecular genetic methods [9]. AVMs were proven to be synthesized through a type-I polyketide synthase (PKS) pathway. Since the sequencing of the complete gene cluster of avermectin by Ikeda, significant progress was achieved in studying biosynthetic and regulatory genes involved in avermectin biosynthesis [10]. The chromosome of *S. avermitilis* was predicted to harbor at least 37 secondary metabolic gene clusters (see the following website: http://avermitilis.ls.kitasato-u.ac.jp/ accessed on 13 November 2022). Among them, there are at least 13 polyketide biosynthetic gene clusters (BGCs), including those for avermectin. In addition to avermectin, oligomycin, filipin, THN (1,3,6,8-tetrahydroxynaphthalene), and an aromatic polyketide were reported to be produced by five polyketide biosynthetic pathways encoded in *S. avermitilis* [10]. A series of large-deletion mutants of *S. avermitilis*, which were constructed by removing nonessential genes and secondary metabolic biosynthetic gene clusters from the wild-type strain, were developed as superhosts for the heterologous production of secondary metabolites [11]. In many cases, the productivities of exogenous metabolites via the heterologous expression of the intact biosynthetic gene cluster in these *S. avermitilis* hosts were significantly improved compared with those of the original producing microorganisms [12]. After sequential deletions of 10 PKS and nonribosomal peptide synthetase biosynthetic gene clusters from *Streptomyces coelicolor*, the expression of the actinorhodin biosynthetic gene cluster in the mutant strain ZM12 resulted in approximately four times as much actinorhodin as its parent *S. coelicolor* strain M145 [13]. Many groups strived to improve doramectin production in recent decades via metabolic engineering or classical mutagenesis. However, the production of doramectin was still quite low; most strains could produce only approximately 1 g/L of doramectin either in flasks or at the bioreactor level [14,15,16]. The strain engineered by Wang could produce 50–60 mg/L of doramectin in flasks [15]. To date, the highest doramectin production was 1068 μg/mL in a 50 L fermenter [14].

Our group previously generated a doramectin overproducer via rational metabolic engineering of an industrial avermectin-producing strain (able to produce ca. 3 g/L avermectin B1a) by knocking out the *bkd* (branched-chain-2-oxo acid dehydrogenase-encoding gene) and *aveD* (5-*O*-methyltransferase-encoding gene) genes along with the replacement of the wild-type *aveC* (spiroketal formation and modification enzyme-encoding gene) with a synthetic *aveC** mutant allele (a variant of the AveC-encoding gene with ten amino acid mutations) [17,18]. In addition, our previous works investigated the influence of the deletion of other PKS BGCs on avermectin production, showing that the termination of *pks3*, *olm*, and *pte* is most beneficial for avermectin overproduction [19]. Additionally, Meng et al. showed that the deletion of the type III polyketide synthase biosynthetic cluster *rpp* can be used to enhance the avermectin production of an industrial overproducer [20]. Because precursors such as malonyl-CoA and methyl-malonyl-CoA are the building blocks of PKS pathways, it is obvious that the competitive relationship among different coexpressed PKSs plays a key role in determining the production of specialized metabolites. All of these results suggest that the removal of competitive PKS gene clusters could direct the PKS biosynthetic flux to the target biosynthetic pathway, which would lead to the overproduction of target compounds.

Exogenous cyclohexanecarboxylic acid was added to be incorporated into the avermectin biosynthetic pathway to produce doramectin. In principle, CoA from precursors, not the free acid, can be efficiently loaded to the PKS assembly line to produce final products. In the doramectin biosynthetic pathway, we proposed that the appropriate CoA ligase would be responsible for the transformation of cyclohexanecarboxylic acid to cyclohexanecarboxylic CoA, which would then be involved in polyketide assembly lines. This process implied that cyclohexanecarboxylic acid activation was probably a key step in limiting doramectin production (Figure 1B).

Higher-production strains are imperative for industrial development and market expansion. In this paper, we discovered an endogenous CoA ligase (FadD17, encoded by SAV_3841) that was probably involved in the activation of cyclohexanecarboxylic acid. As we proposed, doramectin production in the industrial strain was increased significantly by the removal of the main competitive *pks* cluster and the overexpression of the endogenous CoA ligase involved in the cyclohexanecarboxylic acid activation.

## 2. Material and Methods

### 2.1. Bacterial Strains, Plasmids, and Cultivation Conditions

All bacterial strains and plasmids used in this study are listed in Table 1. *Escherichia coli* cells were grown at 37 °C in LB (lysogeny broth) (10 g/L tryptone, 5 g/L yeast extract, 10 g/L NaCl) or on agar supplied with the appropriate antibiotics. *S. avermitilis* strains were sporulated at 30 °C on ISP2 agar (International Streptomyces Project-2 medium containing 10 g/L malt extract, 4 g/L yeast extract, 4 g/L glucose, and 20 g/L agar; pH 7.2 adjusted with 2 N NaOH before autoclaving). MS medium (Mannitol Soya Flour, containing 20 g/L mannitol, 20 g/L soya flour, and 20 g/L agar, prepared with tap water and autoclaved twice with gentle shaking between the two runs) was used for intergeneric conjugation between *E. coli* ET12567 (pUZ8002) and *S. avermitilis*. For fermentation to produce doramectin, *S. avermitilis* strains were incubated on an orbital shaker (shaker type: Forma Thermo Scientific 481 orbital shaker; orbital size: 2.5 cm; manufactured by Thermo Electron Inc., Marietta, OH, USA) at 200 rpm at 28 °C in seed (20 g/L corn starch, 10 g/L defatted soybean flour, 5 g/L glucose, 10 g/L cotton seed flour, pH 7.2) and fermentation (100 g/L corn starch liquefied with thermostable α-amylase, 20 g/L defatted soybean flour, 10 g/L cotton seed flour, 5 g/L yeast extract, NaCl 1 g/L, 2 g/L K_2_HPO_4_, 1 g/L MgSO_4_, 0.8 g/L CHC, 7 g/L CaCO_3_, pH 7.0) medium as described by Xia [18].

### 2.2. DNA Cloning and Sequence Analysis

Alkaline lysis prep of plasmid DNA, transformation of the *E. coli* host, and PCR amplification were performed according to procedures from Sambrook et al. [24]. The protocols for genomic and plasmid DNA isolation and genetic manipulation in *Streptomyces* were described by Kieser et al. [25]. The sequences of oligonucleotide primers for PCR verification and vector construction are described in Table 2. High-fidelity DNA polymerase (FastPfu, TransGen Biotech Inc., Beijing, China) was used for PCRs in ABI9700 Thermocyclers according to the instructions provided by the manufacturer (Thermo Fisher Scientific Inc., Waltham, MA, USA). The reaction programs were initial denaturation at 95 °C for 5 min, followed by 30 cycles of denaturation at 94 °C for 45 s, annealing at 58 °C for 45 s, and extension at 72 °C for 30–120 s (the extension time depended on the length of the target fragments, approximately 2 kb/min). A final extension step was performed at 72 °C for 10 min. The reaction conditions for amplification of the resistance cassette for lambda Red recombination were described in Gust’s reports [21].

### 2.3. Vector Construction and Genetic Manipulation

For the sequential deletion of the biosynthetic gene clusters, first, the genes were replaced by the FRT-*aac(3)IV*-FRT cassette (plasmid pHY773 containing this cassette is presented in Appendix A of the Appendix A). Then, the recombined cosmids were digested with the proper restriction enzyme and cloned into pHY-642 (Appendix A of the Appendix A). Then, the deletion vectors were constructed via cleavage of the *aac(3)IV* gene by the FLP recombinase (the schematic strategy to delete the target cluster is illustrated in Appendix A of the Appendix A). The vector for deletion of the *olm* gene cluster was constructed by cloning the flanking fragments into pHY-642 (Appendix A of the Appendix A). All vectors were introduced into *S. avermitilis* via conjugation. The positive recombinants were verified via PCR amplification and Sanger DNA sequencing (Shanghai Sangon Bioengineering Inc., Shanghai, China).

### 2.4. Analysis of Doramectin Production with High-Performance Liquid Chromatography (HPLC)

Thirteen-day fermentation cultures were extracted with 9 volumes of methanol, followed by centrifugation at 16,000× *g* for 10 min. The liquid phase was passed through a 0.2 μm membrane filter and analyzed with an Agilent 1260 HPLC system (Agilent Inc., Santa Clara, CA, USA) using an Agilent Poroshell 120 Ec-C18 column (2.7 μm, 4.6 × 50 mm). The column was eluted at a flow rate of 1 mL/min for 10 min with acetonitrile:methanol:H_2_O (40:40:20). Metabolites were monitored at a wavelength of 245 nm. The production was calibrated with standard doramectin from Sigma-Aldrich Inc. (Indianapolis, IN, USA).

### 2.5. Statistical Methods

One-way ANOVA was employed to compare the production of different mutant strains. GraphPad Prism version 9.0.0 for Windows (GraphPad Software, San Diego, CA, USA) was used for statistical analysis. The significance of the differences between strains were evaluated using a *t* test.

## 3. Results

### 3.1. Sequential Deletion of the PKS Clusters from the Doramectin-Producing Strain

On the basis of our previous doramectin single component-producing strain DM203, the recombinant strains DM205, DM206, and DM209 were obtained via sequential deletion of the *pks3*, *olm*, and *pte* biosynthetic gene clusters, respectively. The details of the deletion regions are shown in Table 3. All recombinants were verified via PCR amplification with the primers listed in Table 2 and Sanger sequencing of the amplified fragments (Appendix A of the Appendix A). The production of doramectin in these recombinants was investigated via flask fermentation. The production of these strains is presented in Figure 2.

With the deletion of more gene clusters, the doramectin production gradually increased. Strain DM209 (with the deletion of the *pks*3, *olm*, and *pte* clusters) demonstrated the highest production among the tested strains. The doramectin production of DM209 was approximately 60% higher than that of the initial strain DM203.

### 3.2. Verification of the CHC Activation Protein-Encoding Gene

A search of the annotation database of the *S. avermitilis* genome revealed 23 putative CoA ligase-encoding genes (Appendix A) (the search was carried out at http://avermitilis.ls.kitasato-u.ac.jp/ accessed on 13 November 2022). Among them, FadD17 (SAV_3841) was annotated as cyclohex-1-ene-1-carboxylate:CoA ligase, implying that FadD17 is probably involved in the activation of CHC, the precursor of doramectin biosynthesis. Therefore, the FadD17 overexpression vector pSETD17 was constructed via cloning downstream of the constitutively strong promoter *PermE**. The plasmid was incorporated into DM206 to construct the FadD17 overexpression strain DM207. Strain DM215 was constructed as a *fadD17* null mutant derived from strain DM206 with PCR-targeting methods (the schematic strategy to delete the target cluster is illustrated in Appendix A of the Appendix A). The doramectin production of strains DM206, DM207, and DM215 is compared in Figure 3.

The overexpression of FadD17 can enhance the production of doramectin. The deletion of *fadD17* led to a significant decrease in the doramectin production. The doramectin production of strain DM215 was approximately 20% of that in the DM206 strain. This result implies that FadD17 is probably the key CoA ligase involved in the CHC activation for doramectin biosynthesis in *S. avermitilis*.

### 3.3. Construction of a Doramectin Hyperproduction Strain

The above results show that both the deletion of the PKS cluster and the overexpression of FadD17 (SAV_3841) could promote doramectin production. Therefore, the FadD17 overexpression vector was introduced into the DM209 strain (the strain with the deletion of three PKS clusters). The doramectin production of the resulting recombinant strain DM223 was compared with that of the DM203 and DM209 strains (Figure 4).

The doramectin production of DM223 was approximately 723 mg/L, which was approximately 260% higher than that of the initial strain DM203.

## 4. Discussion

The pedigree of the constructed *S. avermitilis* mutant strains, which are derivatives of DM203, is presented in Figure 5. The deletion of the *pks*-3, *olm*, and *pte* biosynthetic gene clusters significantly enhanced the doramectin synthesis in *S. avermitilis* (Figure 2). In fact, we also tried to delete more PKS clusters based on strain DM209. However, all the obtained strains showed no significant enhancement in the production of doramectin. Moreover, some strains showed deficiencies in growth. The *S. coelicolor* strain ZM12, which was obtained via sequential deletions of 10 PKS and nonribosomal peptide synthetase biosynthetic gene clusters in the M145 strain, can produce approximately four times as much actinorhodin as its parent [13]. However, some intermediate strains, such as ZM8 and ZM6, produced less actinorhodin than the parent strain M145. We hypothesized that the deletion of the precursor competition pathway would be helpful in redirecting the secondary metabolic flux. However, the flux directions could be case dependent. The manipulation of the competitive PKS gene cluster can be a potentially effective approach in improving the production of target compounds.

For mutasynthesis, feeding precursor activation plays an important role in determining the production of the final products. In this paper, we show that the proper CoA ligase is involved in the biosynthesis of doramectin. By combining the deletion of competitive gene clusters and the overexpression of precursor activation genes, the production of doramectin was significantly increased in the final strains. All of these results show that cluster deletion and precursor activation was a rational approach for doramectin industrial strain improvements.

Usually, CHC is fed only once during doramectin fermentation in flasks. Higher CHC concentrations in the broth cause growth defects [14,15]. The optimization of the CHC feeding process is probably an option to enhance the production of doramectin. A cyclohexylcarbonyl CoA biosynthetic gene cluster was reported to be engineered to construct a doramectin strain [9,14,26]. Endogenous cyclohexylcarbonyl CoA can probably help solve this dilemma [15,26]. The initial *S. avermitilis* strain can produce approximately 3 g/L of avermectin B1a, which means that the total AVM production could probably be up to 5 g/L. Therefore, much can be done to promote doramectin production. With the development of synthetic biology, the coordination of the biosynthesis of doramectin and CHC-CoA probably facilitates their production.

With the development of synthetic biology in recent years, gene editing technologies such as CRISPR–Cas have begun to be widely applied to engineer *Streptomyces* [27,28,29]. Nevertheless, there are obstacles in engineering industrial strains for low genetic manipulation efficiency. Here, the long flanking arms from the cosmid can facilitate homologous recombination for engineering industrial *S. avermitilis* strains. For targeting deletion regions, CRISPRi approaches could be an efficient way to test target regions [28,29,30].

## Figures and Tables

**Figure 1 bioengineering-10-00739-f001:**
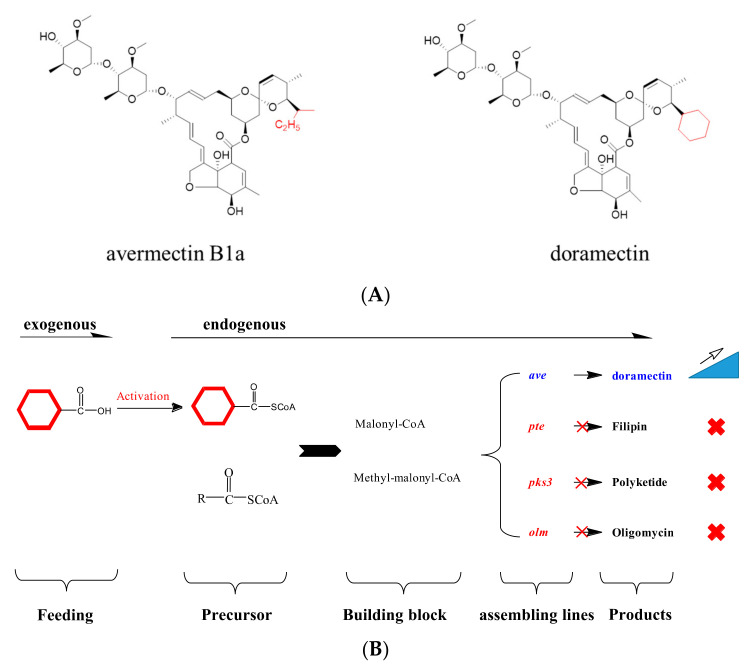
Schematic graph of the strategy to enhance doramectin production in *S. avermitilis*. (**A**) The chemical structures of avermectin B1a and doramectin. The different moieties are shown in red. (**B**) The proposed schematic strategy to improve doramectin production in *S. avermitilis*.

**Figure 2 bioengineering-10-00739-f002:**
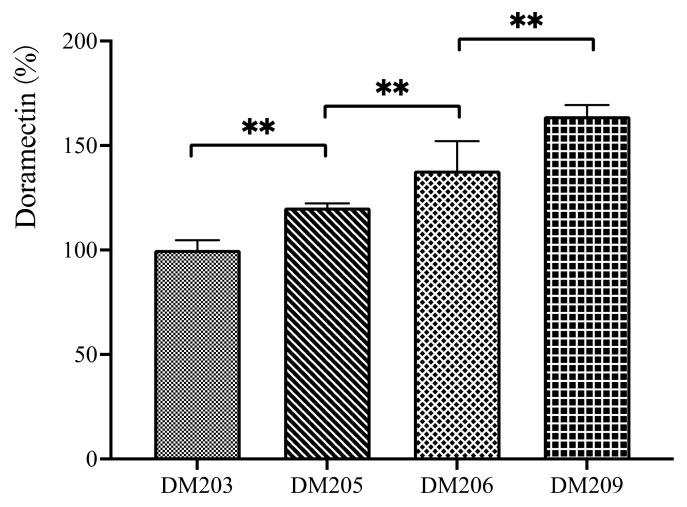
The production of doramectin in the three *pks* null mutant strains. Strains DM205 (∆*pks-3*), DM206 (∆*pks3*-∆*olm*), and DM209 (∆*pks3*-∆*olm*-∆*pte*) are all derived from DM203. Two asterisks denote a statistically significant difference (*p* ≤ 0.01) between the production of two strains.

**Figure 3 bioengineering-10-00739-f003:**
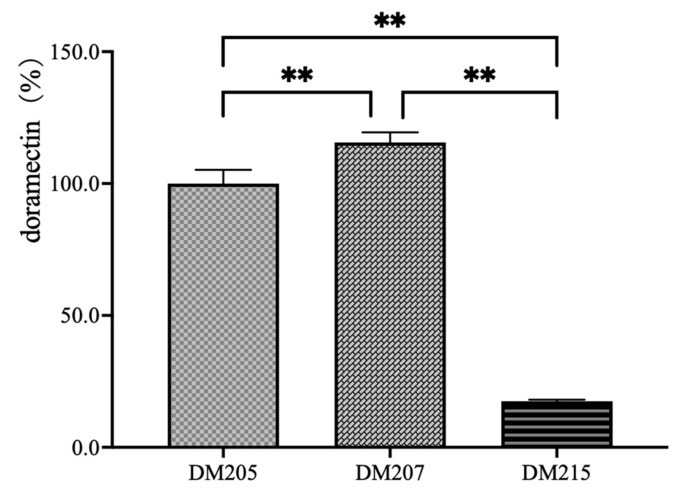
The effect on the production of doramectin by FadD17 compared to the parent strain DM205. DM207 is equivalent to DM205 with a PermE*-driven *fadD17.* DM215 is a *fadD17* mull mutant derived from DM205. Two asterisks denote a statistically significant difference (*p* ≤ 0.01) between the production of two strains.

**Figure 4 bioengineering-10-00739-f004:**
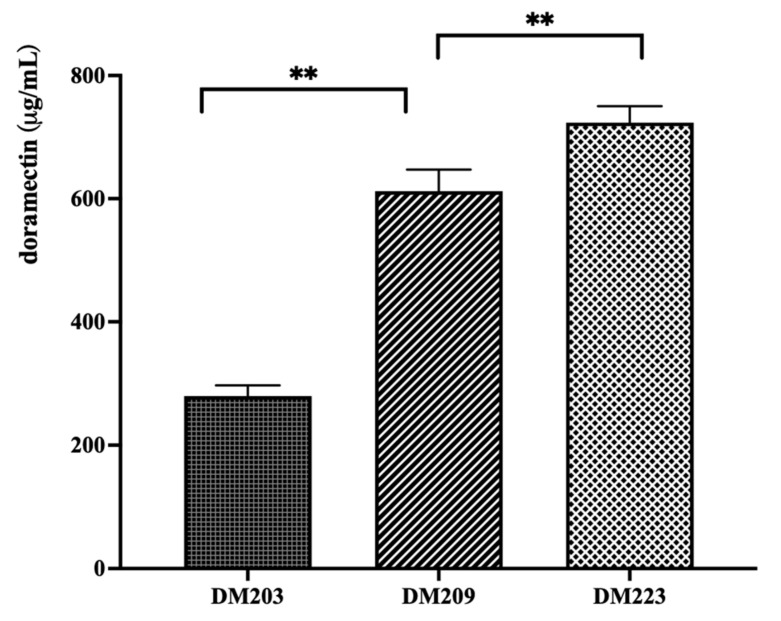
The production of the doramectin hyperproduction strain DM223. Two asterisks denote a statistically significant difference (*p* ≤ 0.01) between the production of two strains.

**Figure 5 bioengineering-10-00739-f005:**
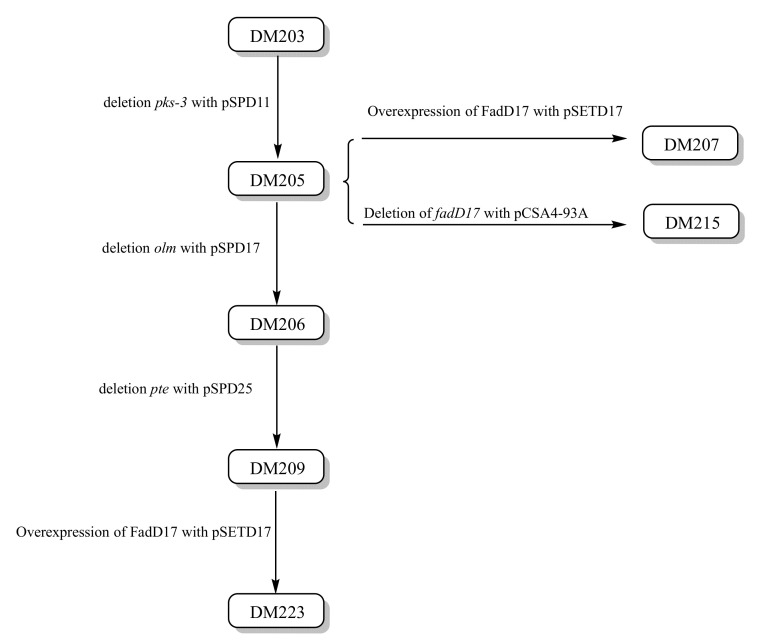
Pedigree of the mutant *S. avermitilis* strains. Steps involved in sequentially deleting the three endogenous secondary metabolite gene clusters.

**Table 1 bioengineering-10-00739-t001:** Plasmids and strains used in this study.

Strain or Plasmids	Genotype/Description	Sources/Reference
*Streptomyces avermitilis*		
DM203	Doramectin-producing strain	[17]
DM205	∆*pks3* mutant of DM203	This study
DM206	∆*pks3-*∆*olm* mutant of DM205	This study
DM209	∆*pks3-*∆*olm*-∆*pte* mutant of DM206	This study
DM207	DM205 with a copy of PermE*-*fadD17*	This study
DM215	∆*fadD17* mutant of DM205	This study
DM223	DM209 with a copy of PermE*-*fadD17*	This study
*Escherichia coli*		
DH5α	Host for general cloning	Invitrogen
BW25113 (pIJ790)	Host containing L-arabinose-inducible lambda Red recombination system	[21]
BT340	Host containing inducible FLP recombination system	[21]
ET12567 (pUZ8002)	*dam dcm hsdM cm kan*, containing a nontransmissible RP4-derived plasmid pUZ8002	[22]
Plasmids		
pIJ773	*bla colE1-ori* FRT-*aac(3)IV-oriT-*FRT cassette	[21]
pHY773	*bla colE1-ori* FRT-*aac(3)IV-*FRT cassette	[23]
pHY642	*Bla colE1-ori* *tsr oriT*	[23]
pSET152	*colE1 ori aac(3)IV oriT phiC31int attP lacZα*	[22]
pCSA4-65	Cosmid contains a c. 40-kb insert (includes the *pks3 g*ene cluster) of *S. avermitilis*	[19]
pCSA11-69	Cosmid contains a c. 40-kb insert (includes the *pteA1* gene) of *S. avermitilis*	[19]
pCSA4-93	Cosmid contains a c. 40-kb insert (includes the *fadD17* gene) of *S. avermitilis*	[19]
pCSA4-65A	*pks3 gene cluster* deletion of pCSA4-65 (PCR-targeting)	This study
pCSA11-69A	*pte* gene cluster deletion of pCSA11-69 (PCR-targeting)	This study
pSPD11	*pks3* gene cluster markerless deletion plasmid	This study
pSPD17	*olm* gene cluster markerless deletion plasmid	This study
pSPD25	*pte* gene cluster markerless deletion plasmid	This study
pSETD17	pSET152 derivative for *PermE*-*driven expression of *fadD17*	This study
pCSA4-93A	*fadD17* gene deletion of pCSA4-93 (PCR-targeting)	This study

**Table 2 bioengineering-10-00739-t002:** Primers used in this study.

Primer	Sequences (5′-3′)	Used for
pks3F	GCAGATCTTGCGCAGGGAGACGAACGCGTCGGGCCCGGCTGTAGGCTGGAGCTGCTTC	Amplification of the FRT-*aac(3)IV*-FRT cassette to target the *pks3* cluster in pCSA4-65 via lambda Red recombination
pks3R	AGGCCCCGGCCGCGGCGCCCTTGGGCCGTCAGCGGGGTGATTCCGGGGATCCGTCGACC	
pks3DF	GACCACCTTCACCTCGTTGC	To verify the deletion and replacement of *pks3* in *S. avermitilis* DM205 and pCSA4-65A, respectively.
pks3DR	CATCAGCCTTTCCGACTTCC	
olmLF	CGG*AAGCTT*TCCTCGAGGGACTCACCGAC	Amplification of the upstream and downstream homologous arms to construct pSPD17.
olmLR	GAG*GGATCC*TGAGAGCGCCTCCAGTTCCC	
olmRF	TGC*GGATCC*CCATAGAACCTTTCGTGCTA	
olmRR	TAC*GAATTC*AACTCGGCGTCCATGTGGAT	
olmDF	ATGACGGGAAGGGCGGAGGTGT	To verify the deletion of the *olm* cluster in *S. avermitilis* DM206
olmDR	GTCGTAGAGGAGGAAGAGCGGTGC	
pteF	CGACGAACGTGGTCACGCCCAGCTCGTGCAGGGTGTGCAATTCCGGGGATCCGTCGACC	Amplification of the FRT-*aac(3)IV*-FRT cassette to target the *pte* cluster in pCSA11-69 via lambda Red recombination
pteR	GCCCACACCCGCGTCGCCAGAAACTCCCGCACCTGCGCCTGTAGGCTGGAGCTGCTTC	
pteDF	AGGGATGCGTCCTGAGTGAGA	To verify the deletion and replacement of *pte* in *S. avermitilis* DM209 and pCSA11-69A, respectively
pteDR	CGGTGAACATTGCGACTGCTT	
fadD17F	GGAATTC*CATATG*GTGAACGACACCGCACACGCGCTCA	Construction of pSETD17 and pHX85D17
fadD17R	GGAATTCCATATG TCACCGCGCATAGCGCTCCCTCAGC	
fadD17-59F	GCGCTCAGCACCTCCGGCACGCTCTGGGAACTCGTCGTCATTCCGGGGATCCGTCGACC	Amplification of the FRT-*aac(3)IV*-*oriT*-FRT cassette to target *fadD17* in pCSA4-93 via lambda Red recombination
fadD17-58R	GATCCTCGATCTCCTTCGCCGAGATGTTCTCGCCCTTGCTGTAGGCTGGAGCTGCTTC	
fadD17-PF	CTGGATACCAGCAGGCTAGTGCC	To verify the deletion and replacement of pks-3 in *S. avermitilis* DM207 and pCSA4-93A, respectively
fadD17-PR	CGCCCGCAGATACGAGGTCAT	

**Table 3 bioengineering-10-00739-t003:** Information on the deleted gene clusters from *S. avermitilis* DM203.

Deleted Clusters	Annotated Region *		Deletion Region *		Length of Deletion Region (bp)
	Start (nt)	End (nt)	Start (nt)	End (nt)	
*pks3*	2,773,878	2,784,841	2,777,920	2,784,838	6918
*olm*	3,534,525	3,634,592	3,534,529	3,634,591	100,062
*pte*	487,415	567,017	546,803	568,027	21,224

* The nt positions refer to the genome sequence of the *S. avermitilis* strain MA-4680, accession No. NC_003155. The *pks3* was deleted in strain DM205. The *pks3* and *olm* were deleted in strain DM206. The *pks3*, *olm*, and *pte* were deleted in strain DM209.

## Data Availability

Data is contained within the article or Appendix A.

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
