# Peer review of "Rationally Improving Doramectin Production in Industrial Streptomyces avermitilis Strains"

_bioengineering, 2023, doi:10.3390/bioengineering10060739_

Round 1
Reviewer 1 Report
(1) This manuscript is poorly organized, lack of locus. Some terms and jargons are not not used, described or explained at all in the text.
(2) For example, Table 1 lists 26 strains and plasmids. Less than 10 of them are actually mentioned (used or described) in the text. The genotype description becomes largely meaningless and useless to the readers. What was the purpose of having all those references of the sources of the plasmids/strains? Do we expect the readers to conduct a literature search to find out the sequence composition and structural features of the list of plasmids and strains??
(3) The same applies to Table 2. What is the purpose of including all the primers without actually describing how they were used and in what plasmid construction step. Of course, none of the construction details were described and there is no way to follow without knowing the sequences of the genes and/or the plasmids. This presentation makes no sense if we don't have the plasmid construct and sequence information.
(4) There are many plasmid/vector/sequence drawing and analysis softwares available, some are free. Why not present a flow chat starting with DM203, showing the steps on how it is engineered to 205, 206 and 209, then to 207 and 215, 223. The deletions of pks-3, olm pte and the cloning of SAV_3841 can be clearly read from the flow diagrams. The production of DM223 is the primary aim and achievement of this work. Why can't we focus on a clear and logical presentation of the cloning process (steps)??
Some specific comments:
(1) Table 3 can be improved by including two more columns: (A) a column showing corresponding strains DM205, 206, 209, (B) a column showing the kb sizes of the detetion regions. For the Table 3 heading: indicate deletions were from DM203.
(2) Figure 2. Move the labels A, B, C and D to the top of the photos/graphs. The current positions at the bottom confuse with the x-axis labels.
(3) Please check to see if the size of the deletion regions (Table 3) match the gel (Figure 2A, B, C. What are the sizes of the bands imply? Describe in the legend or discuss in the text.
(4) Figure 3 legend: Add "compared to the parent clone DM203" to the end.
(5) Figure 2D and 3B use % doramectin (y-axis label). Figure 4 uses mg/L. Explain why the inconsistency.
(6) What is the 1711 bp on the gel? What and how does it tell of DM215?
(7) Please check that names of all microbes/strains are italicized.

Author Response
In this study, the author constructed a doramectin production improved strain by sequential deletion of three competitive PKS gene clusters and overexpression of potential key CoA-ligase coding gene. Eventually doramectin production increased about 2.6 times than that of parental strain. The article is well organized and its result is appropriate and promising, hereby this work is worth being published in Bioengineering after minor revision.
Answer: Thanks a lot for the positive comments on this manuscript.
- The genotype of strains DM215 and DM223 in the article should be listed in table 1.
Answer: We add the information of DM25 and DM223 in this revised manuscript.
- I suggest using fadD17 as the unified gene name instead of using ID SAV_3841 in the article.
Answer: Thanks for your suggestion. We change all the SAV_3841 to fadD17 in this revised manuscript.
- The description of Figure 4 “strain DM215” should be corrected to “strain DM223”.
Answer: I am sorry for this mistake. We corrected this in the revised version
(1) This manuscript is poorly organized, lack of locus. Some terms and jargons are not used, described or explained at all in the text.
Answer: We carefully checked this manuscript and corrected these mistakes.
(2) For example, Table 1 lists 26 strains and plasmids. Less than 10 of them are actually mentioned (used or described) in the text. The genotype description becomes largely meaningless and useless to the readers. What was the purpose of having all those references of the sources of the plasmids/strains? Do we expect the readers to conduct a literature search to find out the sequence composition and structural features of the list of plasmids and strains??
Answer: For this is a short communication manuscript many detail of the experiment procedures haven’t been mentioned. So we present the physical map of plasmids and schematic strategy of deletion in the supplementary materials. The references in this table was listed for the interested readers, which probably helpful for them to use these materials.
(3) The same applies to Table 2. What is the purpose of including all the primers without actually describing how they were used and in what plasmid construction step. Of course, none of the construction details were described and there is no way to follow without knowing the sequences of the genes and/or the plasmids. This presentation makes no sense if we don't have the plasmid construct and sequence information.
Answer: Thanks a lot for this suggestion. We add more description of plasmids construction precedure in the supplementary materials.
(4) There are many plasmid/vector/sequence drawing and analysis softwares available, some are free. Why not present a flow chat starting with DM203, showing the steps on how it is engineered to 205, 206 and 209, then to 207 and 215, 223. The deletions of pks-3, olm pte and the cloning of SAV_3841 can be clearly read from the flow diagrams. The production of DM223 is the primary aim and achievement of this work. Why can't we focus on a clear and logical presentation of the cloning process (steps)??
Answer: We added a pedigree diagram of the mutants S. avermitilis strains in this manuscript. And the physical map of pSPD17 for deletion olm was provided in the supplementary material. This manuscript was to describe the process and strategy for construction of DM223.
Some specific comments:
(1) Table 3 can be improved by including two more columns: (A) a column showing corresponding strains DM205, 206, 209, (B) a column showing the kb sizes of the detetion regions. For the Table 3 heading: indicate deletions were from DM203.
Answer: Thanks for your suggestion. We modified this table on your suggestion. A column to show the sizes of the detetion regions was appended. The deletion of each strains was described in the footnote of Table 3.
(2) Figure 2. Move the labels A, B, C and D to the top of the photos/graphs. The current positions at the bottom confuse with the x-axis labels.
Answer: Thanks a lot for this. For other reviewers suggestion, these A,B and C panels has been moved to supplementary material.
(3) Please check to see if the size of the deletion regions (Table 3) match the gel (Figure 2A, B, C. What are the sizes of the bands imply? Describe in the legend or discuss in the text.
Answer: We added description of these bands in the figure legend and figures in the supplimentary material. For the deletion fragments are too long, expected bands only can be amplified in the mutants.
(4) Figure 3 legend: Add "compared to the parent clone DM203" to the end.
Answer: We add these to the figure legend.
(5) Figure 2D and 3B use % doramectin (y-axis label). Figure 4 uses mg/L. Explain why the inconsistency.
Answer: Usually the production of secondary metabolites fluctuates during different batches of fermentation. Figure 2D and 3B was to compare the productivity of these strains in different steps. For the consistence, we presented these differences by percentage. In figure 4, we need to showed the final production. So production was presented by concentration.
(6) What is the 1711 bp on the gel? What and how does it tell of DM215?
Answer: 1711 bp on this gel was the expected amplification contain an in-frame replacement of fadD17 with the aac(3)IV-oriT cassette. So we add description of this in the figure legend of Figure S5
(7) Please check that names of all microbes/strains are italicized.
Answer: We carefully checked these in this new version.
Reviewer 2 Report
The authors describe another approach of metabolic engineering of avermectin-producing S. avermitilis to improve the production of the compound. Although avermectin (and doramectin) production is studied for decades, the approach described in the manuscript might deserve publication. However, the manuscript has a lot of drawbacks.
Major concerns:
1. The manuscript is a communication, however, the introduction section is extremely long. It should be revised and shortened. The authors describe in detail a lot of general knowledge of avermectin history and production, which should be said in a few sentences. At the same time, authors give only limited details about their own previous results on the topic, which should be given in more detail.
2. Authors should be precise with microbiological and genetic nomenclature, which is used without proper attention. This reviewer highlighted some of the issues in minor concerns, but the manuscript should be carefully read by the authors themselves and the errors should be corrected.
3. Authors should give detailed knockout schemes, including primer binding sites, BGCs, etc. in the ESM. At the same time, PCR genotyping of the obtained mutants should be removed into ESM.
4. Authors should calculate the statistical significance of the obtained results and show it on histograms; more details on statistical parameters should be provided.
5. Since the approach utilized is far not new, authors should compare their approach with other similar works in the discussion section and clearly state the novelty of the obtained results.
6. Finally, the manuscript is poorly written, and contains multiple typos, and English should be revised. Authors should respect the editors and the reviewers and not submit the manuscript with multiple typos, which should be proofread before submission.
Minor concerns:
16-18 – sentence nor clear, requires rewriting: Thus, here we propose a rational strategy to improve doramectin production by the termination of competing polyketide biosynthetic pathways combined with the overexpression of CoA-ligase providing precursors for the polyketide biosynthesis.
19-20 – proven here or elsewhere? Please clarify.
22 – here and further do not forget spaces before units.
23-29 – These sentences are better to be removed: this approach is well known and was described by multiple authors before. Better say shortly: To summarize, our work demonstrated a novel viable approach to creating doramectin overproducers, which might contribute to the cost reduction of this valuable compound in the future.
32 – The introduction is too large and general for a communication. Please rewrite it concisely.
33 –are a series of…
35 – citation format is not proper throughout the manuscript. Please check citation requirements for MDPI Bioengineering and cite literature properly.
57 – proven.
59-60 – enzymatic; characters is a very odd term; better say: significant progress was achieved studying biosynthetic and regulatory genes…
61 – you cannot mine information.
62 – Chromosome of S. avermitilis…
66-67 – references?
67-72 – the sentence is very hard to read, please revise.
84 – However, the production of doramectin was still quite low: most strains could produce only around 1 g/L of doramectin either in flasks or on the bioreactor level…
85 – please compare with the wild type, because 1 g/L is a lot.
87-89 - Our group previously generated a doramectin overproducer via rational metabolic engineering of an industrial avermectin-producing strain (able to produce ca. 3 g/L avermectin B1a) by knocking out bkd and aveD genes along with the replacement of aveC with a synthetic mutant allele aveC*. – the reader gets confused with the genes. Please specify what bkd, aveD, and aveC encode and the difference between aveC and aveC*.
88 – avermectin.
90 – In addition, our previous works investigated the influence of the deletion of other PKS BGCs on avermectin production, showing that the termination of pks3, olm, and pte is most beneficial for avermectin overproduction.
120 – Figure 1. Please revise the formulas, they are very hard to read. I suggest the ACS style. -- Also, assembly line. – better use (A) and (B). – legend should be more informative and self-explanatory – please extend.
125 – Escherichia in full, since first mentioned
126 – Luria Bertani is colloquial, and LB means lysogeny broth.
127 – S. avermitilis – Italics.
127 – ISP2 is self explanatory; also give composition of the medium or cite paper where the composition is mentioned.
128 – same for MS.
129 – ET12567 (pUZ8002) or ET12567 pUZ8002+.
129 - S. avermitilis – Italics!
134 – Table 1. – Revise C31 to φC31.
- Why some genes in strain descriptions are in Italics, while others – not?
148 – Table 2. S. avermitilis – Italics!!
151 – aac(3)IV – italics.
153 - aac(3)IV – italics!
156 – Where was the sequencing performed? Please specify.
162 – developed is an improper term here.
180 – Why figure 2 is separated into two pages?
– Here, A, B, and C panels are not necessary for the main text and should be removed to ESM; for better understanding, A, B, and C panels should be augmented with the scheme of chromosomes with the positions of the deleted BGCs and scheme for knockout approach.
- please give some statistical information on the results shown in D and access probability.
197 – overexpression.
202 – Table 4 is not important for the main text and should be removed from ESM.
204 – Figure 3. Panel A should be removed from ESM.
- also, please give some statistical information on the results shown in B and access probability.
216 – doramectin.
Overall, the manuscript might deserve publication after intensive revision, proofreading, and grammar check.
Author Response
The authors describe another approach of metabolic engineering of avermectin-producing S. avermitilis to improve the production of the compound. Although avermectin (and doramectin) production is studied for decades, the approach described in the manuscript might deserve publication. However, the manuscript has a lot of drawbacks.
Answer: Thanks a lot for your valuable suggestion. We tried a lot to improve this manuscript.
Major concerns:
- The manuscript is a communication, however, the introduction section is extremely long. It should be revised and shortened. The authors describe in detail a lot of general knowledge of avermectin history and production, which should be said in a few sentences. At the same time, authors give only limited details about their own previous results on the topic, which should be given in more detail.
Answer: We rephrased and shorten the introduction part. We hope it would be suitable for publication.
- Authors should be precise with microbiological and genetic nomenclature, which is used without proper attention. This reviewer highlighted some of the issues in minor concerns, but the manuscript should be carefully read by the authors themselves and the errors should be corrected.
Answer: thanks a lot for this. We carefully check the whole manuscript and corrected these errors.
- Authors should give detailed knockout schemes, including primer binding sites, BGCs, etc. in the ESM. At the same time, PCR genotyping of the obtained mutants should be removed into ESM.
Answer: we appended more detail about mutants construction in ESM including the cluster, primer binding sites. All PCR genotyping gel graph was moved to ESM.
- Authors should calculate the statistical significance of the obtained results and show it on histograms; more details on statistical parameters should be provided.
Answer: We added statistic information in this new version. For comparing productions of different strains, one way ANOVA approach was chosen to calculate statistical significance.
- Since the approach utilized is far not new, authors should compare their approach with other similar works in the discussion section and clearly state the novelty of the obtained results.
Answer: We included more discussion about this in this version. The novelty of this work was about the generation of higher production doramectin strain. The approach is practcal for industrial strains.
- Finally, the manuscript is poorly written, and contains multiple typos, and English should be revised. Authors should respect the editors and the reviewers and not submit the manuscript with multiple typos, which should be proofread before submission.
Answer: We are sorry for the poor writing of this manuscript. We carefully checked the typos and grammar. This new version improved a lot. If necessary, English editing service would be arranged by MDPI.
Minor concerns:
16-18 – sentence nor clear, requires rewriting: Thus, here we propose a rational strategy to improve doramectin production by the termination of competing polyketide biosynthetic pathways combined with the overexpression of CoA-ligase providing precursors for the polyketide biosynthesis.
Answer: We corrected this sentence according your suggestion.
19-20 – proven here or elsewhere? Please clarify.
Answer: We proved the role of this CoA ligase here.
22 – here and further do not forget spaces before units.
Answer: We are very sorry for these mistakes. We carefully check the whole manuscript and corrected them.
23-29 – These sentences are better to be removed: this approach is well known and was described by multiple authors before. Better say shortly: To summarize, our work demonstrated a novel viable approach to creating doramectin overproducers, which might contribute to the cost reduction of this valuable compound in the future.
Answer: We revised this following your suggestion.
32 – The introduction is too large and general for a communication. Please rewrite it concisely.
Answer: Thanks a lot for your comments. We rewrote this part in this revised version.
33 –are a series of…
Answer: We corrected this in the revised version.
35 – citation format is not proper throughout the manuscript. Please check citation requirements for MDPI Bioengineering and cite literature properly.
Answer: We formatted the citation style of this manuscript following instruction of this Journal.
57 – proven.
Answer: We corrected the typo.
59-60 – enzymatic; characters is a very odd term; better say: significant progress was achieved studying biosynthetic and regulatory genes…
Answer: We rephrase this sentence.
61 – you cannot mine information.
Answer: We changed “information” to “genes”
62 – Chromosome of S. avermitilis…
Answer: We put the “Chromosome of” before “S. avermitilis”
66-67 – references?
Answer: We insert references here.
67-72 – the sentence is very hard to read, please revise.
Answer: We revised the sentence to “A series of large-deletion mutants of S. avermitilis, which were constructed by removing non-essential genes and secondary metabolic biosynthetic gene clusters from wild type strain, were developed as hosts for the heterologous production of secondary metabolites”
84 – However, the production of doramectin was still quite low: most strains could produce only around 1 g/L of doramectin either in flasks or on the bioreactor level…
Answer: We corrected this sentence following your advice.
85 – please compare with the wild type, because 1 g/L is a lot.
Answer: Though 1 g/L is a lot for academic research. But for commercial research, it is still low.
87-89 - Our group previously generated a doramectin overproducer via rational metabolic engineering of an industrial avermectin-producing strain (able to produce ca. 3 g/L avermectin B1a) by knocking out bkd and aveD genes along with the replacement of aveC with a synthetic mutant allele aveC*. – the reader gets confused with the genes. Please specify what bkd, aveD, and aveC encode and the difference between aveC and aveC*.
Answer:We corrected this and appended information to these genes in this revised version.
88 – avermectin.
Answer:This typo was correct in this new version
90 – In addition, our previous works investigated the influence of the deletion of other PKS BGCs on avermectin production, showing that the termination of pks3, olm, and pte is most beneficial for avermectin overproduction.
Answer: Thanks a lot. We revised these sentence according your suggestion.
120 – Figure 1. Please revise the formulas, they are very hard to read. I suggest the ACS style. -- Also, assembly line. – better use (A) and (B). – legend should be more informative and self-explanatory – please extend.
Answer: We changed the formulas to ACS style. And corrected the “assembly line”. More information was appended in the legend.
125 – Escherichia in full, since first mentioned
Answer: We revised this following your suggestion.
126 – Luria Bertani is colloquial, and LB means lysogeny broth.
Answer: We referred this to “Sezonov G, Joseleau-Petit D, D'Ari R. Escherichia coli physiology in Luria-Bertani broth. J Bacteriol. 2007 Dec;189(23):8746-9. doi: 10.1128/JB.01368-07.” We keep “Luria-Bertani” here and append compositon of this medium. The "LB" was removed.
127 – S. avermitilis – Italics.
Answer: We revised this following your suggestion. And carfully checked the whole manuscript.
127 – ISP2 is self explanatory; also give composition of the medium or cite paper where the composition is mentioned.
Answer: We explained the ISP2 here and provided the composition of the medium.
128 – same for MS.
Answer: We explained the MS here and provided the composition of the medium.
129 – ET12567 (pUZ8002) or ET12567 pUZ8002+.
Answer: We revised this following your suggestion.
129 - S. avermitilis – Italics!
Answer: We revised this following your suggestion.
134 – Table 1. – Revise C31 to φC31.
- Why some genes in strain descriptions are in Italics, while others – not?
Answer: We are sorry for that. We revised this following your suggestion.
148 – Table 2. S. avermitilis – Italics!!
Answer: We revised this following your suggestion.
151 – aac(3)IV – italics.
Answer: We revised this following your suggestion.
153 - aac(3)IV – italics!
Answer: We revised this following your suggestion.
156 – Where was the sequencing performed? Please specify.
Answer: The amplification products was sequenced by Shanghai Sangon Bio-engineering Inc.
162 – developed is an improper term here.
Answer: We changed “developed” by “eluted”.
180 – Why figure 2 is separated into two pages?
– Here, A, B, and C panels are not necessary for the main text and should be removed to ESM; for better understanding, A, B, and C panels should be augmented with the scheme of chromosomes with the positions of the deleted BGCs and scheme for knockout approach.
Answer: We moved A,B and C panels to ESM. And combined with the procedure to construt these recombinant strains.
- please give some statistical information on the results shown in D and access probability.
Answer: We add statistical analysis information to this graph.
197 – overexpression.
Answer: We revised this following your suggestion.
202 – Table 4 is not important for the main text and should be removed from ESM.
Answer: We revised this following your suggestion. The Table 4 was moved to ESM and named as Table S1.
204 – Figure 3. Panel A should be removed from ESM.
- also, please give some statistical information on the results shown in B and access probability.
Answer: We revised this following your suggestion. Panel A moved to ESM. And statistical information on the results was appended to the legend.
216 – doramectin.
Answer: We corrected this typo.
Overall, the manuscript might deserve publication after intensive revision, proofreading, and grammar check.
Answer: We carefully checked the whole manuscript and the grammar. We think it is suitable for publication.
Reviewer 3 Report
In this study, the author constructed a doramectin production improved strain by sequential deletion of three competitive PKS gene clusters and overexpression of potential key CoA-ligase coding gene. Eventually doramectin production increased about 2.6 times than that of parental strain. The article is well organized and its result is appropriate and promising, hereby this work is worth being published in Bioengineering after minor revision.
1. The genotype of strains DM215 and DM223 in the article should be listed in table 1.
2. I suggest using fadD17 as the unified gene name instead of using ID SAV_3841 in the article.
3. The description of Figure 4 “strain DM215” should be corrected to “strain DM223”.
Author Response
In this study, the author constructed a doramectin production improved strain by sequential deletion of three competitive PKS gene clusters and overexpression of potential key CoA-ligase coding gene. Eventually doramectin production increased about 2.6 times than that of parental strain. The article is well organized and its result is appropriate and promising, hereby this work is worth being published in Bioengineering after minor revision.
Answer: Thanks for your positive evaluation on our manuscript.
- The genotype of strains DM215 and DM223 in the article should be listed in table 1.
Answer:We added genotype information of these two strain in Table 1.
- I suggest using fadD17 as the unified gene name instead of using ID SAV_3841 in the article.
Answer: Thanks for the suggestion. We replaced the “SAV_3841” with fadD17 (gene) and FadD17 (protein).
- The description of Figure 4 “strain DM215” should be corrected to “strain DM223”.
Answer: We are sorry for this mistake. We corrected this in the new version.
Round 2
Reviewer 2 Report
The authors put a certain effort into improving the manuscript. However, multiple grammatical and other mistakes remain in the text, hampering the comprehension of the manuscript. The same goes for ESM. This reviewer is afraid that this manuscript could not be published without using an English editing service from MDPI. Some of the typos and other issues are given below:
37 – derivative
41 – 5-O-methyltransferase
42-45 – references?
53 – clusters (see website…
56 – THN (1,3,6,8-tetrahydroxynaphthalene) – check spaces before and after brackets everywhere in the manuscript!
56-57 - have been reported to be produced
57 - S. avermitilis [10].
70 – when the wild type produced how much?
113 – let us refer to Giuseppe Bertani himself, shall we? doi: 10.1128/JB.186.3.595-600.2004. No such thing as Luria-Bertani broth exists, LB means lysogeny broth.
167 – remove the comma before The
222 – I do not see asterisks in Figure 4.
260 – biology in.
261 – there are.
Author Response
Dear Reviewer,
Thanks a lot for your help. Please see the attachment.
Best regards,
Haiyang

Round 3
Reviewer 2 Report
After rounds of rebuilding, this manuscript has undergone significant improvement and absolutely deserves publication in current form.